# A new method for customized fetal growth reference percentiles

**Katherine L. Grantz**[1]*, **Stefanie N. Hinkle**[2], **Dian He**[1,3], **John Owen**[4], **Daniel Skupski**[5], **Cuilin Zhang**[1,6], **Anindya Roy**[7]

1 Division of Population Health Research, Division of Intramural Research, *Eunice Kennedy Shriver* National Institute of Child Health and Human Development, National Institutes of Health, Bethesda, Maryland, United States of America, 2 Department of Biostatistics, Epidemiology and Informatics, Perelman School of Medicine, University of Pennsylvania, Philadelphia, Pennsylvania, United States of America, 3 The Prospective Group, Inc., Fairfax, Virginia, United States of America, 4 Division of Maternal and Fetal Medicine, Department of Obstetrics and Gynecology, Heersink School of Medicine, Birmingham, Alabama, United States of America, 5 Department of Obstetrics and Gynecology, Division of Maternal Fetal Medicine, Weill Cornell Medicine and New York Presbyterian Queens, New York, New York, United States of America, 6 Yong Loo Lin School of Medicine, National University of Singapore, Singapore, Singapore, 7 Department of Mathematics and Statistics, University of Maryland Baltimore County, Baltimore, Maryland, United States of America

* Katherine.grantz@mail.nih.gov

**Data Availability Statement:** Data and accompanying files for the NICHD Fetal Growth Study are available through NICHD BRADS, https://brads.nichd.nih.gov/. The Consortium on Safe

## Abstract

### Background

Customized fetal growth charts assume birthweight at term to be normally distributed across the population with a constant coefficient of variation at earlier gestational ages. Thus, standard deviation used for computing percentiles (e.g., 10th, 90th) is assumed to be proportional to the customized mean, although this assumption has never been formally tested.

### Methods

In a secondary analysis of NICHD Fetal Growth Studies-Singletons (12 U.S. sites, 2009–2013) using longitudinal sonographic biometric data (n = 2288 pregnancies), we investigated the assumptions of normality and constant coefficient of variation by examining behavior of the mean and standard deviation, computed following the Gardosi method. We then created a more flexible model that *customizes both mean and standard deviation* using heteroscedastic regression and calculated customized percentiles directly using quantile regression, with an application in a separate study of 102, 012 deliveries, 37–41 weeks.

### Results

Analysis of term optimal birthweight challenged assumptions of proportionality and that values were normally distributed: at different mean birthweight values, standard deviation did not change linearly with mean birthweight and the percentile computed with the normality assumption deviated from empirical percentiles. Composite neonatal morbidity and mortality rates in relation to birthweight < 10th were higher for heteroscedastic and quantile models (10.3% and 10.0%, respectively) than the Gardosi model (7.2%), although prediction performance was similar among all three (c-statistic 0.52–0.53).

Labor study data are publicly available through the NICHD Data and Specimen Hub (DASH) https://dash.nichd.nih.gov/.

**Funding:** This research was supported, in part, by the Division of Population Health Research, Division of Intramural Research, *Eunice Kennedy Shriver* National Institute of Child Health and Human Development (NICHD), National Institutes of Health; and, in part, with Federal funds for the NICHD Fetal Growth Studies – Singletons (Contract Numbers: HHSN275200800013C; HHSN275200800002I; HHSN27500006; HHSN275200800003IC; HHSN275200800014C; HHSN275200800012C; HHSN275200800028C; HHSN275201000009C); and in part, with Federal funds for the Consortium on Safe Labor (Contract Number HHSN267200603425C). KLG has contributed to this work as part of her official duties as an employee of the United States Federal Government. Intramural investigators designed the study that was implemented by clinical site investigators. For the present analysis, the funder had no role in study design, data analysis, or preparation of the manuscript. All manuscripts undergo clearance before submission for publishing.

**Competing interests:** The authors have declared that no competing interests exist.

## Conclusions

Our findings question normality and constant coefficient of variation assumptions of the Gardosi customization method. A heteroscedastic model captures unstable variance in customization characteristics which may improve detection of abnormal growth percentiles.

## Trial registration

**ClinicalTrials.gov identifier:** NCT00912132.

## Introduction

Fetal undergrowth as often characterized by fetal growth restriction (FGR) and small-for-gestational age (SGA) is associated with an increased risk of perinatal morbidity and mortality [1]. SGA is often defined as birthweight $< 10^{th}$ percentile using a population based growth reference [2]. However, this approach identifies fetuses who are constitutionally small but otherwise healthy and misses fetuses who did not meet their growth potential but whose weight is at or above the $10^{th}$ percentile. In 1992, Gardosi et al proposed a customized method for birthweight references that took into account six pregnancy characteristics known to influence birthweight and thought to be physiologic, namely gestational age, maternal pre-pregnancy weight, height, race, parity, and fetal sex [3]. This method was further extrapolated from birthweight to estimated fetal weight during gestation by using fetal ultrasonographic biometric data and a commonly used fetal growth reference from Hadlock [4, 5]. The percentiles for the ultrasound estimated fetal weight (EFW) curves (e.g., $10^{th}$, $50^{th}$ and $90^{th}$) were proportionately adjusted upwards or downwards according to the Gardosi method's expected optimal birthweight at term for a given set of maternal and fetal characteristics. Customized fetal growth references are appealing as they provide a more personalized definition of FGR and SGA, in line with a precision medicine approach; however, whether their use improves the clinical detection of fetuses with suboptimal growth and at risk of morbidity and mortality is controversial [6–8]. Nevertheless, they have been recommended for use by national guidelines in some countries including Britain, Ireland and New Zealand [9]. A recent randomized trial did not demonstrate improved prenatal detection of SGA using the Growth Assessment Protocol based on customized fetal growth charts compared to standard care, although the negative results have been questioned because of lack of adherence to the intervention study arm and bleeding of some parts of the intervention in the "standard care" arm [10, 11].

The primary metric of the Gardosi method is a customized term optimal birthweight (TOW) at 40 weeks which is then extrapolated to EFW at any gestational time using the proportionality model [12]. Based on the model and the proportionality assumption, the percentiles (e.g., $5^{th}$, $10^{th}$, $90^{th}$, $95^{th}$ etc.) for the EFW are produced at all gestational ages between 24 and 42 weeks. However, the customized TOW percentiles are based on the assumptions that the distribution of birthweight is normal, and the standard deviation used for calculating the percentiles (e.g., $10^{th}$, $90^{th}$), is proportional to the mean, i.e., the coefficient of variation (CV) is constant; these assumptions have never been formally tested yet have important clinical implications, because different percentile cutoffs will identify different proportions of fetuses as SGA versus non-SGA. This differential classification would potentially increase the risk of stillbirth in those pregnancies where SGA goes undetected or cause unnecessary iatrogenic earlier delivery in pregnancies where SGA is erroneously diagnosed.

This study was a secondary analysis of the NICHD Fetal Growth Studies–Singletons, a prospective pregnancy cohort study with the primary aim to establish fetal growth standards for size and velocity in the U.S. [13–15]. Our objectives were first, to evaluate the assumptions of the Gardosi customization model that the distribution of TOW around its customized mean value was normal and the standard deviation used for calculating the CV was proportional to the mean TOW. Second, we created a new customization method that has more flexibility in calculating customized percentiles using a heteroscedastic regression that customizes *both* mean TOW (and hence EFW by extrapolation) *and* standard deviation [16]. To be precise, the heteroscedastic model customizes a transformed value of the standard deviation but because that makes the standard deviation depend on the customizing factors, hereafter we refer to it as a model for customizing the standard deviation. Also, since clinical outcomes of SGA and LGA are essentially percentiles (e.g. 10[th], 90[th]), we further customized fetal growth using quantile regression, which directly calculates the percentiles without being reliant on the model for the mean and the assumption of normality [17]. We compared the performance of all three customization methods in relation to SGA and LGA birthweight with neonatal morbidity and mortality within the NICHD Fetal Growth Study and also in a concurrent analysis from the Consortium on Safe Labor because it has a larger number of births.

## Materials and methods

### Study design and participants

The NICHD Fetal Growth Studies–Singletons recruited 2334 non-obese women (BMI 19·0–29·9 kg/m²) from four different race/ethnic groups who were non-smokers and had low-risk medical and obstetrical histories (e.g., no chronic diseases) from 2009 to 2013 at 12 U.S. centers. Details of recruitment and study design have been previously reported [18]. An additional 468 women with BMI 30·0–44·9 kg/m² were recruited with similar inclusion criteria, although relaxed to allow certain chronic conditions (e.g. chronic hypertension controlled on medication), given the higher prevalence of concurrent morbidities with obesity [19]. Institutional review board approval was obtained at all participating sites as well as the NIH (IRB approval #09-CH-N152) on December 2009 prior to the study beginning. All participants provided written informed consent prior to data collection.

### Procedures

Gestational age was based on a certain last menstrual period and confirmed by first trimester ultrasound [18]. At enrollment, information on demographics, obstetrical and medical histories, and lifestyle and health leading up to and during the first trimester of pregnancy was collected via in-person interview. After an enrollment sonogram at 10–13 weeks of gestation, women were randomly assigned to one of four ultrasound schedules for follow up visits at ranges 16–22, 24–29, 30–33, 34–37 and 38–41 weeks of gestation. For the assigned study visit, ± 1 week was allowed to accommodate women's availability. Sonographers for the study underwent uniform, centralized training and credentialing. A standardized protocol was used to obtain ultrasound measurements for fetal biometry including head circumference (HC), abdominal circumference (AC), and femur length (FL) at each study visit using identical, high-resolution ultrasound units at each center. The HC, AC, and FL were used to calculate EFW using a Hadlock formula [20]. Information on lifestyle, reproductive and medical history were obtained via in-person interviews at each research visit. Demographic data and antenatal, labor, delivery and neonatal course and outcomes were abstracted from the prenatal record, labor and delivery summary, hospital and neonatal records by trained research personnel. Paternal height and weight were by maternal report.

Outcomes included SGA and LGA birthweight defined as $< 10^{th}$ or $> 90^{th}$ using the Duryea reference with neonatal morbidity and mortality [21]. Neonatal morbidities associated with SGA or LGA included: metabolic acidosis (pH <7.1 and base deficit >12mmol/L), neonatal intensive care unit (NICU) stay greater than three days, pneumonia, respiratory distress syndrome, persistent pulmonary hypertension, seizures, hyperbilirubinemia requiring exchange transfusion, intrapartum aspiration (meconium, amniotic fluid, blood), neonatal death, mechanical ventilation at term, necrotizing enterocolitis, hypoglycemia, hypoxic ischemic encephalopathy, periventricular leukomalacia (SGA only), sepsis based on blood culture (SGA only), bronchopulmonary dysplasia/chronic lung disease (SGA only), retinopathy of prematurity (SGA only), and birth injury (LGA only) [22–26].

SGA and LGA associated with neonatal morbidities were defined similarly in a concurrent analysis of n = 102, 012 deliveries between 37–41 weeks from the Consortium on Safe Labor (CSL) [27]. Pregnancy, labor and delivery information was electronically abstracted from maternal records. Neonatal records included information on gestational age, NICU admission, medical conditions and discharge diagnoses. International Classification of Diseases, 9th Revision, Clinical Modification (ICD-9-CM) codes were collected and linked to deliveries. Outcomes were defined to be consistent with previous CSL studies [28].

## Statistical analysis

Demographic data were summarized as n (%) or mean (± SD). We developed a fetal growth percentile customization model using the Gardosi method [4]. Linear regression was used to predict birthweight at 40 weeks as the outcome, designated as the term optimal weight (TOW), using six customization variables: gestational age, maternal pre-pregnancy weight, height, race/ethnicity, parity, and infant sex. We then explored some of the assumptions of the Gardosi model, namely the assumption of normality and the constant CV assumption for the TOW distribution. If the Gardosi assumption of normality and constant CV were to hold, the percentiles computed based on Gardosi model should agree with the empirical percentiles across different levels of the mean birthweight. We stratified the estimated birthweight into eight contiguous intervals (depicting eight different values of mean birthweight) and investigated the agreement of the empirical percentiles with those obtained from the Gardosi model for each interval. To verify the assumption of constant CV, we looked at the relationship between the empirical standard deviation and the mean birthweight across the different birthweight intervals. As an extension to the Gardosi model which assumes that the standard deviation is proportional to the customized mean, we then created a model to *customize both mean and standard deviation* of the TOW using heteroscedastic regression with predicted birthweight at 40 weeks as the outcome and the same six customization variables [4]. The new customized mean and SD yielded customized values for the target percentiles using the quantile formula for normal distribution (S1 Table).

In a third customization model, we calculated customized percentiles directly using quantile regression with monotonic smoothing, a flexible model that does not assume a normal distribution [17]. Note that quantile regression customizes the target percentiles directly without using a 2-step model where first the customized mean and the customized standard deviation are obtained and then the percentiles are computed using the quantile formula for normal distribution. All three models included the same customizing variables containing cubic and quadratic terms of deviation of gestational time at delivery from the optimal 280 days mark a priori per the Gardosi model. In addition to the six proposed "physiological" variables that influence fetal growth, models also included "pathological" variables, smoking, BMI (kg/m$^{2)}$, and gestational diabetes, gestational hypertensive disease/preeclampsia, and antepartum bleeding. The analysis was centered on 280 days' gestation, height 163 cm, pre-pregnancy weight 64

kg, nulliparous, and Non-Hispanic White race/ethnicity. However, only the coefficients for the six "physiologic" variables (as designated by the Gardosi method) were included in an additive model to calculate the TOW percentiles [12]. The six variables were categorized similar to the Gardosi model with some slight alterations due to the availability of the data. Specifically, we included four race/ethnic groups (Asian, Hispanic, Non-Hispanic Black, Non-Hispanic White) instead of ethnic origin which was not available in our study. Parity 2 and greater (P2+) was combined into one group because of sparse data for higher parity whereas the Gardosi model includes each one separately: P0 (ref), P1, P2, P3, P > = 4. Standard goodness-of-fit and model diagnostics were performed.

The customization method of fetal growth based on the previously noted 6 maternal and fetal factors calculates the term optimal weight at 40 weeks which is then extrapolated back to ultrasound EFW across gestation using the Hadlock reference, proportionately adjusting the percentiles (e.g., 10th, 50th, 90th) upward or downward based on the profile. Therefore, to check cross-sectional consistency of the variance, the heteroscedastic model was executed a second time using EFW for pairs of weeks, i.e., 21–22, 22–23, etc. instead of extrapolating. Pairs of weeks were chosen because there were insufficient observations at each individual week.

Both the heteroscedastic regression model with separately customizable mean and standard deviation and the quantile regression model that explicitly produced customized percentiles were then compared to the Gardosi model [29]. Note that under the normality assumption in the Gardosi and heteroscedastic models, the mean is equal to the median value. We computed the 5th, 10th, 50th, 90th and 95th percentiles for birthweight for deliveries at 37–41 weeks for a hypothetical mother whose customization factors were set to population average values in the NICHD Fetal Growth Studies–Singletons. The analysis was performed for each of the 3 models and the estimated percentiles were plotted for comparison. The equations to calculate the percentiles for the 3 models are presented in the Supplement. We also calculated the mean, median, SD, 10th, and 90th percentiles for the 3 models using EFW (instead of birthweight) at 38 and 39 weeks in the NICHD Fetal Growth Studies–Singletons.

We also compared the performance of the three customization models (Gardosi, heteroscedastic and quantile regression) and the Duryea birthweight reference in relation to SGA and LGA birthweight with neonatal morbidity and mortality [21]. Sensitivity, specificity, positive predictive values (PPV) and negative predictive values (NPV) were calculated for the association between each of the SGA and LGA classifications from the three customization models against the observed neonatal morbidity and mortality using multivariable logistic regression. Comparison of the performance of the customization models was first performed using the EFW at 38–39 weeks from the NICHD Fetal Growth Studies–Singletons. Analyses were then repeated using birthweight from the CSL study (because EFW was not available in the CSL). This step was for examining reproducibility and generalizability of the findings albeit using birthweight, since the CSL study included a much larger sample of deliveries on which out-of-sample prediction performance was tested. Moreover, the NICHD Fetal Growth Studies–Singletons targeted recruitment of low-risk pregnancies whose primary goal was developing a fetal growth standard, excluding pregnancies at higher risk for fetal growth abnormalities; the recruitment criterion for CSL did not have this restriction/limitation.

All analyses were completed with the use of SAS software (version 9·4, SAS Institute, Inc., Cary, NC) or R (version 3·5·2, available at http://www.R-project.org).

## Results

Of the 2802 women recruited for the NICHD Fetal Growth Studies–Singletons, we excluded those who were deemed ineligible after enrollment, fetal anomalies, neonatal aneuploidy,

deactivated (e.g., for pregnancy loss, moved, pregnancy termination, or lost to follow-up), delivered < 37 weeks, or had missing information, leaving 2288 for final analysis (S1 Fig). Study participants were racially/ethnically diverse with a mean maternal age of 28.2 (± 5.4) years; 46% were nulliparous, 56% had a BMI 18.5 to < 25 kg/m$^2$, 26% had a BMI 25 to < 30 kg/m$^2$ and 16% a BMI 30.0 or greater kg/m$^2$ (Table 1).

## Evaluation of customization assumptions

In order to evaluate the assumptions of normality and constant CV we examined the data as follows. The data were sorted by the mean estimated TOW based on the Gardosi model and divided into eight contiguous equal length intervals, where each interval represents cases with a specific value of TOW (the mean birth weight value in the interval). The number of observations in each interval were not equal with fewer observations for the extreme intervals. However, there were substantial observations in each for the mean, the standard deviation and the percentiles to be estimated accurately. For each interval, we computed the empirical percentiles of birthweight and the standard deviation as well as the mean predicted TOW from the Gardosi model. We also computed the percentiles using the normality and the constant CV assumption from the Gardosi model. The results are presented in Figs 1 and 2. Fig 1 shows the relationship between the empirical percentiles of birthweight and those estimated based on Gardosi assumptions for different values of mean birthweight. The empirical percentiles often differ from those obtained from the model. The 5$^{th}$ percentile was generally being over-estimated by the model while the 95$^{th}$ was generally underestimated. The difference was as big as 150gm. In Fig 2, the standard deviation of TOW for specific values of mean TOW are presented. If the Gardosi assumption of constant CV was satisfied the standard deviations would fall on the line with constant slope equal to the value of the CV. However, we observed considerable departure from the constant CV model.

We further investigated the assumption of normality by checking the residuals from the model fits for the Gardosi and the heteroscedastic models. The quantile regression does not assume normality and hence it was not included in the investigation. The residuals did not show any glaring departure from normality (S2 Fig).

## Creation and comparison of three customization models

Table 2 presents the results from the 3 models. As expected, the term optimal weight of 3510 g was similar for both Gardosi and heteroscedastic models since the mean would be the same as the median under the assumption of normality. However, in the quantile regression, the median term optimal weight was lower, 3487 g, challenging the assumption of normality.

The beta-coefficients and standard errors for the mean characteristics in the heteroscedastic model and the Gardosi models were similar (Table 2). Interestingly, only the linear terms for maternal height and weight were statistically significant (in both models) but not the quadratic or cubic terms. However, we retained the quadratic and cubic terms in the model since they are included in the Gardosi model, and our main interest was to assess the variance terms. In the heteroscedastic model, only pre-pregnancy weight significantly affected the standard deviation (linear term β = 0.0145). Some of the other variables showed a potentially non-constant influence on the variability of TOW. Standard goodness-of-fit and model diagnostics indicated that overall, all 3 models appeared to fit well whereas the residuals did not show any appreciable departure.

## Evaluation of model performance across gestation

The heteroscedastic model was executed again using EFW for pairs of weeks, i.e., 21–22, 22–23, etc. instead of birthweight to check the cross-sectional consistency of variance (i.e., whether

**Table 1. Participant and pregnancy characteristics in the NICHD Fetal Growth Studies–Singletons (N = 2,288).**

| Characteristic | n (%) or mean ± SD |
| --- | --- |
| Maternal age (y) | 28.2 (5.4) |
| Maternal height (cm) | 162.8 (7.0) |
| Maternal weight (kg) | 67.4 (14.9) |
| BMI (kg/m$^2$) | |
| < 18.5 | 12 (0.5%) |
| 18.5 to < 25 | 1290 (56.4%) |
| 25 to < 30 | 595 (26.0%) |
| 30 to <35 | 251 (11.0%) |
| ≥ 35 | 140 (6.1%) |
| Parity | |
| 0 | 1062 (46.4%) |
| 1 | 791 (34.6%) |
| 2+ | 435 (19.0%) |
| Race/ethnic group | |
| Non-Hispanic white | 643 (28.1%) |
| Non-Hispanic black | 607 (26.5%) |
| Hispanic | 665 (29.1%) |
| Asian/Pacific Islander | 373 (16.3%) |
| Smoking | 10 (0.4%) |
| Gestational Diabetes | 96 (4.2%) |
| Maternal hypertensive disease | |
| No Hypertension | 2147 (93.8%) |
| Mild Gestational Hypertension | 59 (2.6%) |
| Severe Gestational Hypertension | 4 (0.2%) |
| Mild Preeclampsia | 48 (2.1%) |
| Severe Preeclampsia | 13 (0.6%) |
| Unspecified Hypertension | 17 (0.7%) |
| Other Diseases | |
| Asthma | 5 (0.2%) |
| Thyroid disease | 4 (0.2%) |
| Hematologic disorders | 5 (0.2%) |
| Antepartum bleeding | 620 (27.1%) |
| Abruption | 9 (0.4%) |
| Gestational age at delivery (wk) | 39.5 (1.1) |
| Infant birthweight (g) | 3390.6 (438.5) |
| Infant sex | |
| Male | 1162 (50.8%) |
| Female | 1126 (49.2%) |
| Neonatal death | 1 (0.0%) |
| Paternal height (cm)[a] | 177.66 (8.50) |
| Paternal weight (kg) [a] | 84.8 (16.2) |

[a] n = 2040

Data are from the NICHD Fetal Growth Studies–Singleton.

the model assumptions hold at any unspecified point in gestation, not just at delivery with birthweight) (S2 Table). Though sporadic differences in variances were observed by maternal

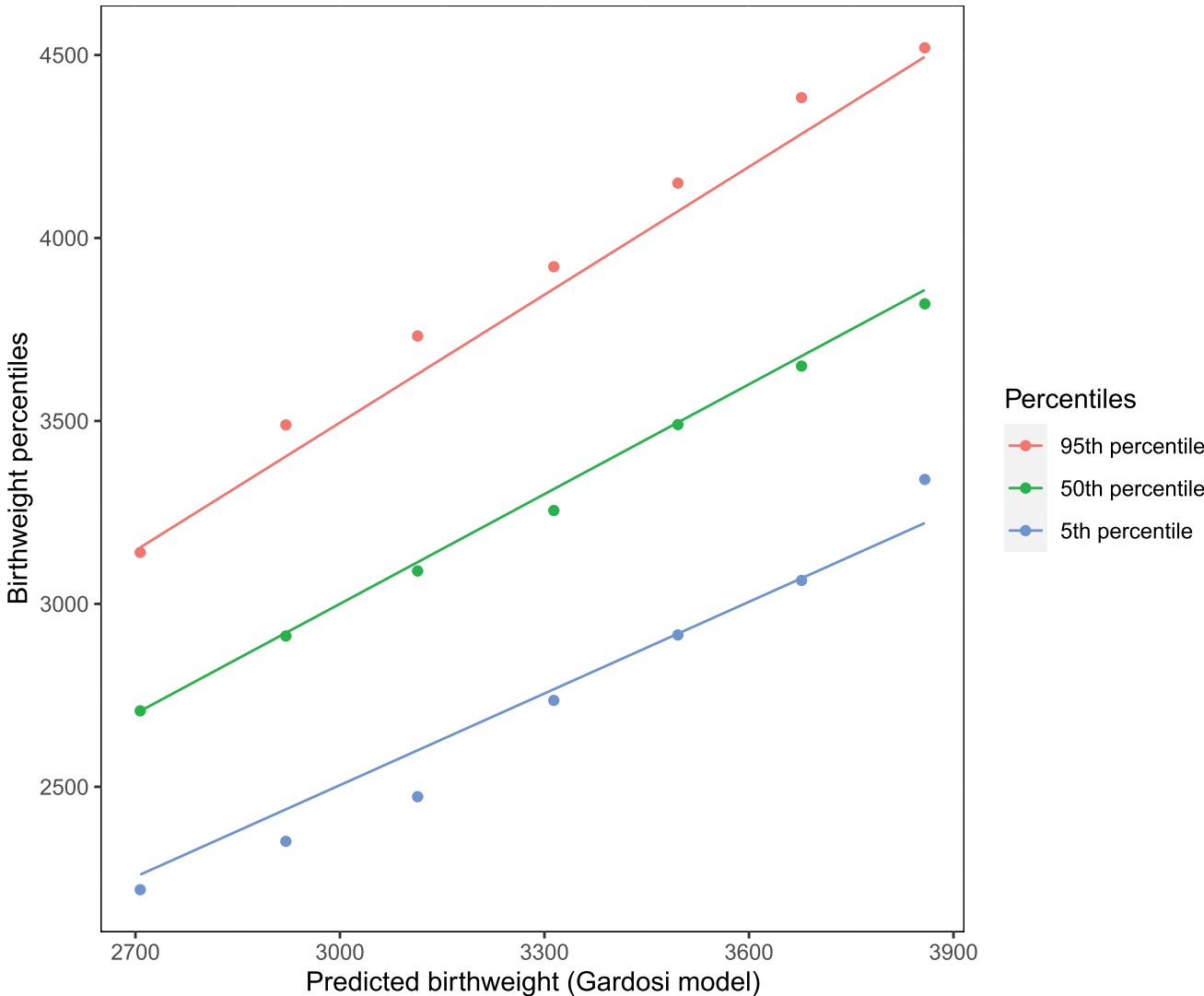

**Fig 1. Empirical percentiles (5ᵗʰ and 95ᵗʰ) of term optimal birthweight at different levels of mean birthweight are compared with the percentiles obtained from the Gardosi model which assumes normality and constant CV to compute the percentiles.** Data are from the NICHD Fetal Growth Studies–Singleton. The lines are from the models while the points are empirical observations.

weight and height, no systematic dependence on any particular characteristic was found across gestation. These findings from the rolling weekly pair analysis indicate that there was no specific departure from the heteroscedastic customization model across gestation. Interestingly, however, the main effects of three of the six characteristics, maternal height, weight, and parity, in mean customization model for EFW were not consistent across gestational weeks. Maternal height was associated with increased EFW from around 28 to 30 weeks of gestation, and again around 33 weeks onward. Maternal weight also was associated with increased EFW from around 29 to 31 weeks and again around 33 weeks onward. Increasing parity was associated with increased EFW starting at the beginning of the third trimester around 28 weeks, although did not reach statistical significance until towards the end of pregnancy (not adjusted for multiple testing).

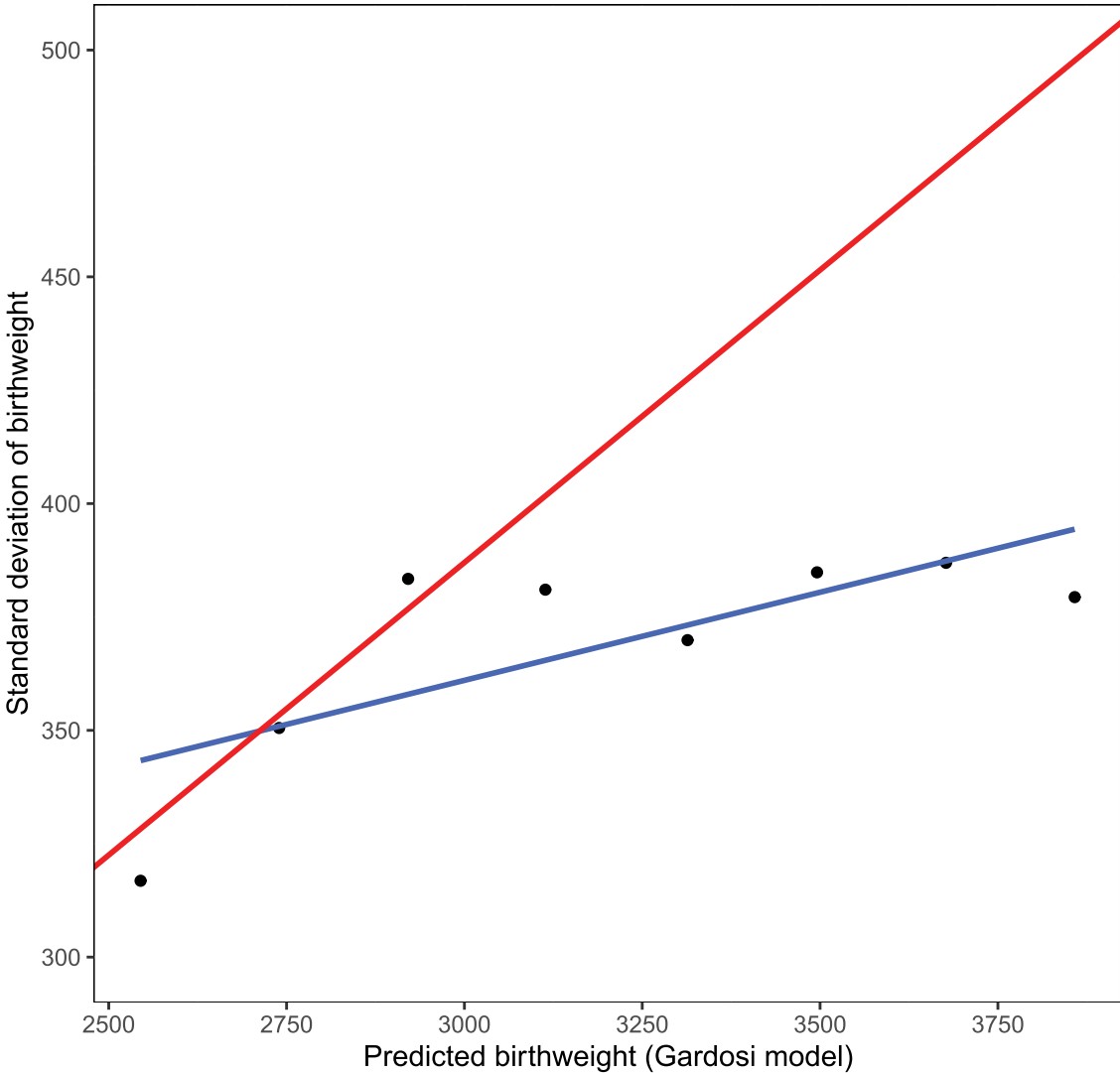

**Fig 2. Empirical and Gardosi model based standard deviation of birthweight at different levels of mean birthweight.** The constant coefficient variation assumption in Gardosi model would imply the empirical standard deviations would fall along a straight line (red line) with constant CV value as its slope. However, the empirical based fitted line (blue line) differs from the line based on the Gardosi assumption and the empirical points don't show a linear pattern. Data are from the NICHD Fetal Growth Studies–Singleton. The sample size of each group is 5, 25, 85, 292, 668, 822, 350, and 41 for the point representing the grouped bins from smallest to largest predicted birthweight, respectively.

### Evaluation of model performance across customization characteristics

Comparison of the 5th, 10th, 50th, 90th and 95th percentiles among the three models were performed for the six characteristics to evaluate model performance. Analyses comparing birthweight for deliveries at 37–41 weeks in the NICHD Fetal Growth Studies–Singletons are presented in Fig 3 for illustration. The 50th percentile was similar across all 3 models with the quantile regression percentile being only slightly lower than the other two. The percentiles for the Gardosi model were father apart than the other two models, meaning that there was a slightly lower birthweight for the 5th and 10th percentile cutpoints and slightly higher birthweight for the 90th and 95th percentile cutpoints than the heteroscedastic and quantile

**Table 2. Coefficients for three customization models in the NICHD Fetal Growth Studies–Singletons (N = 2,288).**

| Variable | Gardosi Model [a] | | | Heteroscedastic Model | | | Quantile Regression Model | | | | |
|---|---|---|---|---|---|---|---|---|---|---|---|
| | Estimate | SE | P | Estimate | SE | P | Estimate 10% | Estimate 50% | Estimate 90% | SE 50% | P 50% |
| *Mean Model* | | | | | | | | | | | |
| Intercept (Term Optimal Weight) | 3509.722 | 21.250 | < .0001 | 3510.000 | 21.264 | < .0001 | 3069.076 | 3486.616 | 4041.833 | 24.289 | < .0001 |
| Gestational age (from 280 d) | | | | | | | | | | | |
| Linear term | 14.944 | 1.818 | < .0001 | 14.944 | 1.850 | < .0001 | 13.366 | 17.147 | 8.282 | 2.269 | < .0001 |
| Quadratic term | -0.282 | 0.191 | 0.1403 | -0.293 | 0.184 | 0.1103 | -0.389 | -0.403 | -0.269 | 0.206 | 0.0509 |
| Cubic term | 0.026 | 0.012 | 0.034 | 0.025 | 0.012 | 0.0379 | 0.027 | 0.008 | 0.042 | 0.016 | 0.6414 |
| Sex | | | | | | | | | | | |
| Male | 67.048 | 7.911 | < .0001 | 67.048 | 8.002 | < .0001 | 55.752 | 66.499 | 71.076 | 9.517 | < .0001 |
| Female | -67.048 | 7.911 | < .0001 | -67.048 | 8.002 | < .0001 | -55.752 | -66.499 | -71.076 | 9.517 | < .0001 |
| Maternal height (from 163 cm) | | | | | | | | | | | |
| Linear term | 5.864 | 1.903 | 0.0021 | 5.861 | 2.007 | 0.0035 | 9.412 | 5.306 | -4.789 | 2.129 | 0.0127 |
| Quadratic term | 0.041 | 0.117 | 0.7252 | 0.034 | 0.114 | 0.7691 | 0.173 | 0.130 | -0.081 | 0.112 | 0.2464 |
| Cubic term | -0.001 | 0.008 | 0.8934 | 0.000 | 0.009 | 0.986 | -0.018 | 0.002 | 0.020 | 0.008 | 0.7861 |
| Maternal prepregnancy weight (from 64 kg) | | | | | | | | | | | |
| Linear term | 7.720 | 1.140 | < .0001 | 7.721 | 1.170 | < .0001 | 4.751 | 7.395 | 13.533 | 1.348 | < .0001 |
| Quadratic term | -0.114 | 0.063 | 0.0706 | -0.119 | 0.065 | 0.0669 | -0.086 | -0.157 | -0.203 | 0.075 | 0.0372 |
| Cubic term | 0.000 | 0.001 | 0.6579 | 0.000 | 0.001 | 0.7709 | -0.001 | 0.001 | -0.001 | 0.001 | 0.5762 |
| Race | | | | | | | | | | | |
| Non-Hispanic black | -189.435 | 21.695 | < .0001 | -189.435 | 21.880 | < .0001 | -182.776 | -193.922 | -225.116 | 24.380 | < .0001 |
| Hispanic | -54.454 | 22.150 | 0.014 | -54.454 | 22.964 | 0.0177 | -77.925 | -65.586 | -74.850 | 28.232 | 0.0203 |
| Asian/Pacific Islander | -49.903 | 26.053 | 0.0556 | -49.903 | 25.876 | 0.0538 | -9.154 | -69.425 | -79.243 | 36.557 | 0.0577 |
| Parity | | | | | | | | | | | |
| 1 | 92.614 | 18.070 | < .0001 | 92.614 | 18.394 | < .0001 | 72.759 | 97.666 | 59.634 | 22.623 | < .0001 |
| 2+ | 101.536 | 22.391 | < .0001 | 101.536 | 23.535 | < .0001 | 117.866 | 102.512 | 45.780 | 28.195 | 0.0003 |
| *Variance Model* [b] | | | | | | | | | | | |
| Intercept | | | | 374.598 | 14.777 | < .0001 | | | | | |
| Gestational age (from 280 d) | | | | | | | | | | | |
| Linear term | | | | -0.004 | 0.007 | 0.604 | | | | | |
| Quadratic term | | | | 0.000 | 0.001 | 0.8844 | | | | | |
| Cubic term | | | | 0.000 | 0.000 | 0.9483 | | | | | |
| Sex | | | | | | | | | | | |
| Male | | | | 0.025 | 0.030 | 0.4035 | | | | | |
| Female | | | | -0.025 | 0.030 | 0.4035 | | | | | |
| Maternal height (from 163 cm) | | | | | | | | | | | |
| Linear term | | | | -0.012 | 0.007 | 0.0922 | | | | | |
| Quadratic term | | | | 0.000 | 0.000 | 0.3882 | | | | | |
| Cubic term | | | | 0.000 | 0.000 | 0.1498 | | | | | |
| Maternal prepregnancy weight (from 64 kg) | | | | | | | | | | | |

(*Continued*)

**Table 2.** (Continued)

| Variable | Gardosi Model [a] | | | Heteroscedastic Model | | | Quantile Regression Model | | | | |
|---|---|---|---|---|---|---|---|---|---|---|---|
| | Estimate | SE | P | Estimate | SE | P | Estimate 10% | Estimate 50% | Estimate 90% | SE 50% | P 50% |
| Linear term | | | | 0.015 | 0.004 | < .0001 | | | | | |
| Quadratic term | | | | 0.000 | 0.000 | 0.6577 | | | | | |
| Cubic term | | | | 0.000 | 0.000 | 0.5303 | | | | | |
| Race | | | | | | | | | | | |
| Non-Hispanic black | | | | -0.111 | 0.082 | 0.1783 | | | | | |
| Hispanic | | | | -0.011 | 0.082 | 0.8963 | | | | | |
| Asian/Pacific Islander | | | | -0.041 | 0.100 | 0.6829 | | | | | |
| Parity | | | | | | | | | | | |
| 1 | | | | 0.039 | 0.070 | 0.5723 | | | | | |
| 2+ | | | | 0.040 | 0.083 | 0.6278 | | | | | |

Note: 0.000 is used for any value <0.001.

[a] All three models included the same customizing variables containing cubic and quadratic terms of deviation of gestational time at delivery from the optimal 280 days mark a priori per the Gardosi model [12]. In addition to the six proposed "physiological" variables (as designated by the Gardosi method) that influence fetal growth, models also included smoking, BMI (kg/m2), gestational diabetes, gestational hypertensive disease/preeclampsia, and antepartum bleeding. Analysis was centered on 280 days' gestation, height 163 cm, prepregnancy weight 64 kg, nulliparous, and Non-Hispanic White race/ethnicity. However, only the coefficients for the six "physiologic" variables were included in an additive model to calculate the term optimal weight percentiles.

[b] Variance is only for the heteroscedastic model.

regression models which were more aligned. In the heteroscedastic and quantile regression models, EFW 10th and 90th percentiles were also closer to one another than the Gardosi model across a range of maternal weights: 57kg, 64kg and 75kg for the 25th, 50th, and 75th percentiles, respectively (Table 3). For example, EFW 10th percentile at 37–38 weeks for a woman with a pre-pregnancy weight of 57 kg was 99g larger with customized variance (2571g heteroscedastic) and 131 g larger for quantile regression (2603 g) vs. Gardosi (2472 g), while EFWs at the 90th percentile were 99 g and 26 g smaller, respectively.

## Assessment of the effect of paternal characteristics on birthweight

Paternal height and weight were also independently associated with birthweight (S3 Table). In general, for each cm increase in paternal height from the average 177.8 cm, there was an approximately 3 g increase in EFW (4 g using quantile regression), compared to the 5 g increase (4 g increase for quantile regression) in EFW for each cm increase in maternal height from the average 163 cm when both were included in the model. For each 1 kg increase in paternal weight from the 81.6 kg average, there was also a 3 g increase in EFW compared to the 7 g increase in EFW for each kg increase in maternal weight from the average of 64 kg.

## Summary of actual and predicted birthweight for the NICHD Fetal Growth Studies-Singletons

To evaluate comparative model performance, we calculated the median, 10th, 90th percentiles for birthweight in the NICHD Fetal Growth Study. The empiric (observational) mean birthweight (37–41 weeks) was 3371 g which was similar to the estimated term optimal birthweight of 3374 g for the Gardosi model and 3375 g for the heteroscedastic model, indicating that these models performed well at estimating observed mean birthweight. The estimated term optimal birthweight from the quantile regression was 3350 g, which was 22 g lower than the observed

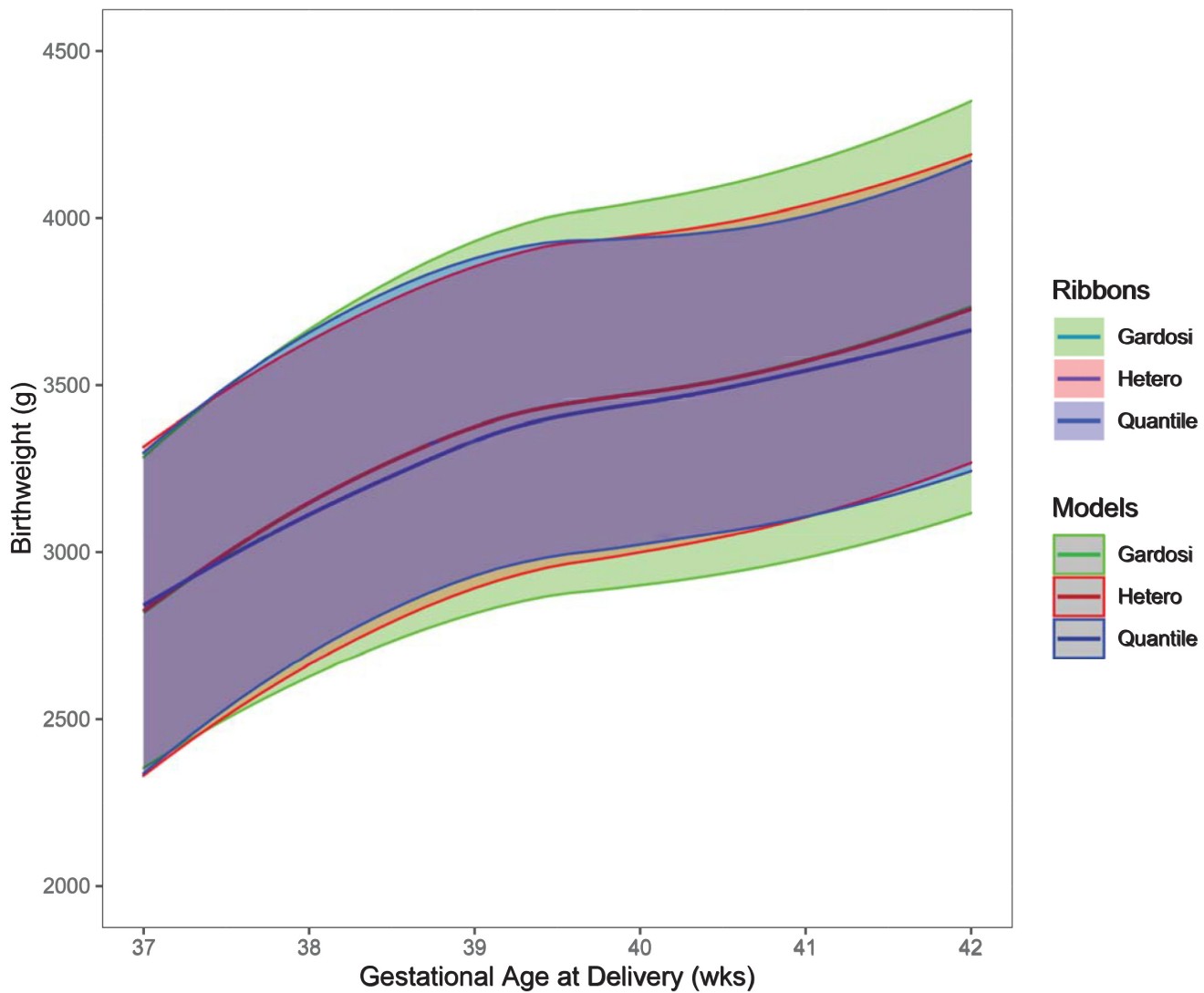

**Fig 3. Comparison of the three customization models for birthweight at term.** The 5th, 10th, 50th, 90th and 95th percentiles for the Gardosi, heteroscedastic and quantile regression models for birthweight for deliveries at 37–41 weeks. Data are from the NICHD Fetal Growth Studies–Singleton.

birthweight and expected since the quantile regression is modeling the median of the distribution rather than the mean. The standard deviations of the predicted birthweights were narrower for the customization models (248 g for Gardosi, 246 g for the heteroscedastic, and 234 g for the quantile regression) compared to 447 g for the observed birthweight. This difference can be explained because the extremes of the observed birthweight distribution are more widely dispersed than those of the predicted distributions. Such a phenomenon is not unexpected. Since the customization models use measures of central tendency (i.e., mean/median), the predicted distributions of birthweights are well-aligned with the observed distribution at the center of the data. The discrepancy in the percentiles between the observed distributions and the predicted distributions are more pronounced toward the tail of the distribution, with the 10th and the 90th percentiles differing by two to three hundred grams.

**Table 3. Comparison of three different methods across different levels of maternal prepregnancy weight for birthweight at 37–42 weeks in the NICHD Fetal Growth Studies–Singletons (N = 2,288).**

| | 25th percentile maternal weight | 50th percentile maternal weight | 75th percentile maternal weight |
|---|---|---|---|
| Prepregnancy weight–kg | 56.7 | 63.5 | 74.8 |
| Gardosi 90th Percentile–g | 4017 | 4085 | 4170 |
| Heteroscedastic 90th Percentile–g | 3901 | 3984 | 4094 |
| Quantile regression 90th Percentile–g | 3932 | 4035 | 4164 |
| Gardosi Term Optimal Weight–g | 3447 | 3506 | 3579 |
| Heteroscedastic Term Optimal Weight–g | 3447 | 3506 | 3579 |
| Quantile regression Term Optimal Weight–g | 3424 | 3483 | 3549 |
| Gardosi 10th Percentile–g | 2878 | 2927 | 2988 |
| Heteroscedastic 10th Percentile–g | 2994 | 3028 | 3065 |
| Quantile regression 10th Percentile–g | 3030 | 3067 | 3109 |

Note: All calculations were performed in SAS with non-rounded numbers. Proc GLM was used for the Gardosi based models, Proc AUTOREG was used for the heteroscedastic models and Proc QUANTREG was used for the quantile regression model. The percentiles were calculated per the equations in S1 Table. For example, the 90th percentile from the Gardosi model was calculated as term optimal weight (TOW) = TOW + (1.28 * Sigma), where sigma = TOW*0.129, and where .129 is the coefficient of variation for the study population; the 90th percentile for the heteroscedastic model was calculated as TOW = TOW + (1.28*customized sigma), where the customized sigma was determined using the customized variance beta coefficients: 354.504 for the 25th maternal weight percentile, 373.2451 for the 50th maternal weight percentile, 402.0685 for the 75th maternal weight percentile. For example, for the 25th maternal weight percentile

epsi_het25 = 374.5975 *exp(0.5*(0.0145*(-7.301)-0.000103*(-7.301*-7.301)-0.0000028*(-7.301*-7.301*-7.301))

-0.0123*(0) -0.000357*(0)+0.0000472*(0)

-0.003582*(0) -0.000112*(0) -0.000003257*(0)

+ 0*(-0.1109)+ 0*(-0.0107)+ 0*(-0.0407)+ 0*(0.0394)+ 0*(0.0403)

+ 0.0236*(0)))

In both the Gardosi and Heteroscedastic, TOW was specific to each maternal weight percentile. For the quantile regression, the 10th and 90th percentiles were calculated directly from the model.

### Neonatal morbidity prediction across three customization models

Finally, we applied the models to birthweight data at 37–41 weeks in the CSL and compared classification of SGA and LGA in relation to neonatal morbidity and mortality in the CSL (Table 4). While the composite neonatal morbidity and mortality rates in relation to SGA were higher for the heteroscedastic and quantile regression models (10.3% and 10.0%, respectively) than the Gardosi model (7.2%), the prediction performance was similar among the 3 customization models as well as the Duryea population-based birthweight reference (c-statistic 0.52–0.54) [21]. The pattern was similar for LGA (c-statistic 0.53 for all). Findings were similar in the NICHD Fetal Growth Studies–Singletons analysis for EFW at 38–39 weeks (S4 Table).

### Discussion

We performed an in-depth examination of the statistical assumptions of the Gardosi customization method [4]. Our investigation indicates that the standard deviation varies differently than the mean birthweight across gestation for the six customization characteristics. These findings question the constant coefficient of variation assumption of the Gardosi customization model that the standard deviation, and therefore the customized percentiles, is proportional to the mean birthweight. Therefore, we created a model to simultaneously estimate *both* customized mean *and* standard deviation with heteroscedastic regression. Also, since our findings questioned the assumption that the data were normally distributed, we further investigated direct customization using a quantile regression model that does not assume normal distribution. While 50th percentile EFW was similar across models, 10th and 90th percentiles

**Table 4. Comparison of model performance for the three different methods in detecting SGA and LGA with morbidity for birthweight at 37–42 weeks of gestation in the Consortium on Safe Labor study (N = 102,012).**

| Classification | n | Composite neonatal morbidity % | PPV | NPV | Sensitivity | Specificity | Odds ratio (95% CI) | c-statistic (95% CI) |
|---|---|---|---|---|---|---|---|---|
| **LGA > 90th** | | | | | | | | |
| **Duryea** | 9,895 | 9.7 | 7.7 | 95.1 | 14.5 | 90.6 | 1.62 (1.50–1.75) | 0.53 (0.52–0.53) |
| **Gardosi** | 7,814 | 7.7 | 8.5 | 95.1 | 12.6 | 92.6 | 1.80 (1.65–1.96) | 0.53 (0.52–0.53) |
| **Heteroscedastic** | 10,497 | 10.3 | 7.7 | 95.1 | 15.3 | 90.0 | 1.63 (1.50–1.76) | 0.53 (0.52–0.53) |
| **Quantile** | 10,211 | 10.0 | 8.0 | 95.1 | 15.4 | 90.3 | 1.69 (1.56–1.83) | 0.53 (0.52–0.53) |
| **SGA < 10th** | | | | | | | | |
| **Duryea** | 9,385 | 9.2 | 6.5 | 96.3 | 15.2 | 91.04 | 1.82 (1.66–1.99) | 0.54 (0.53–0.54) |
| **Gardosi** | 7,294 | 7.2 | 7.2 | 96.3 | 13.2 | 93.09 | 2.05 (1.86–2.25) | 0.53 (0.52–0.53) |
| **Heteroscedastic** | 10,538 | 10.3 | 5.6 | 96.3 | 14.9 | 89.85 | 1.55 (1.41–1.69) | 0.52 (0.52–0.52) |
| **Quantile** | 10,201 | 10.0 | 5.9 | 96.3 | 15.0 | 90.20 | 1.63 (1.49–1.78 | 0.53 (0.52–0.53) |

Note: In the Consortium on Safe Labor Study, birthweight was predicted by using models' coefficients from the NICHD Fetal Growth Studies. Large- and small-for-gestational-age (LGA and SGA, respectively) were defined according to different models, then calculated the positive predictive value (PPV), negative predictive value (NPV), sensitivity, specificity, odds ratio and c-statistic using neonatal morbidity as the outcome. Neonatal morbidities were selected specifically for SGA or LGA based on increased risks associated with these and included: metabolic acidosis (pH <7.1 and base deficit >12mmol/L), NICU stay greater than three days, pneumonia, respiratory distress syndrome, persistent pulmonary hypertension, seizures, hyperbilirubinemia requiring exchange transfusion, intrapartum aspiration (meconium, amniotic fluid, blood), neonatal death, mechanical ventilation at term, necrotizing enterocolitis, hypoglycemia, hypoxic ischemic encephalopathy, periventricular leukomalacia (SGA only), sepsis based on blood culture (SGA only), bronchopulmonary dysplasia/chronic lung disease (SGA only), retinopathy of prematurity (SGA only), and birth injury (LGA only) [22–26].

for the Gardosi model were father apart, resulting in lower birthweight for 10th percentile and higher for 90th percentile cutpoints, than other two models. Composite neonatal morbidity and mortality rates in relation to birthweight < 10th percentile was higher for the heteroscedastic and quantile regression models (10.3% and 10.0%, respectively) than the Gardosi model (7.2%), although prediction performance was similar among all three (c-statistic 0.52–0.53). Thus, while there was some departure from the assumptions of the Gardosi model, it still performed well in comparison to a more flexible heteroscedastic model. While quantile regression resolves the issue about assumption of normality, its similar performance in estimating the percentiles indicates that other two models may generally be robust with respect to the assumption of normality, at least for the study population considered, since the effect of non-normality did not have an appreciable impact on model performance. In summary, the heteroscedastic model is equally straightforward to implement as the Gardosi model and has the advantage of being able to capture unstable variance in the customization characteristics if needed.

The quantile regression model seems to be a natural choice for modeling quantiles when standard assumptions of normal distribution models are suspect. Quantile regression was used to create the WHO fetal growth charts and also for a customized fetal growth reference in an African-American population [30, 31]. However, the price of the greater flexibility of the quantile regression is that it generally requires a greater sample size to yield accuracy as comparable to the linear regression models [17]. In the study by Kabiri et al., a customized fetal growth reference based on quantile regression did not improve prediction of perinatal morbidity compared with ultrasound references [30]. While in the present cohort the model also did not show significant improvement in terms of birthweight prediction, it is expected that as more data from controlled studies become available, the merits of flexible models compared to linear regression-based models could be better evaluated in the context of birthweight customization.

Our investigation into the statistical assumptions of customization methods of proportional standard deviation across birthweight values is novel. In addition, the effect of the covariates on fetal growth across gestation had also been assumed to be fixed, but we found the effect of pre-pregnancy weight on EFW was both nonconstant and non-linear, and in the heteroscedastic model, maternal pre-pregnancy weight significantly affected the variance. Maternal height and parity were also associated with increased EFW starting at the beginning of the third trimester, with little influence in the first and second trimesters. Some of the other customization variables showed some non-constant influence on the distribution of EFW, although these findings were not statistically significant which could have been due to limited power. Also, the quadratic or cubic terms for maternal height and weight were not statistically significant in either the Gardosi or heteroscedastic model, indicating that a linear term may be sufficient. Removal of maternal weight from the customization model has previously been found to identify a greater proportion of LGA neonates with deliveries complicated by shoulder dystocia, NICU admission and neonatal respiratory problems that were not identified by a population based definition of LGA, although that analysis used the outcome of birthweight [32]. These findings indicate that the characteristics (i.e. maternal weight) and terms in the Gardosi customization model (i.e. quadratic and cubic) that are currently included may be unnecessary. In our analysis of EFW at 38–39 weeks, customization with the heteroscedastic model identified a slightly higher proportion of SGA neonates with morbidity (8.9%) compared to the Gardosi method (5.7%), with a similar pattern for LGA and SGA neonates $< 5^{th}$ percentiles. Perhaps the ability of the heteroscedastic model to allow for unstable variance in the customization characteristics yielded a slight incremental improvement. Therefore, the heteroscedastic customization method has potential to identify more fetuses at risk of growth restriction and macrosomia, with associated improvement in targeting antenatal surveillance and obstetric intervention to reduce neonatal morbidity and stillbirths.

Paternal factors have not traditionally been included in customization charts. We found that increasing paternal height and weight had a positive, independent influence on fetal growth, although maternal height and weight had a stronger effect. These findings are similar to findings from the Generation R cohort of EFW in the Netherlands [33] and fetal biometric measurements in an Italian cohort [34] although another study from the UK also found maternal weight to have a stronger influence on birthweight, while maternal and paternal height had similar contributions [35]. The fact that maternal factors have a stronger influence on anthropometrics during fetal life compared to paternal factors has been hypothesized to be due to maternal preservation in conditions of constraint [36].

While the six customization characteristics (gestational age, maternal pre-pregnancy weight, height, race, parity, infant sex) are known to influence fetal growth, it is unclear whether the changes in fetal growth in relation to these characteristics are a normal physiologic adaptation or associated with increased risk for perinatal morbidity and mortality. Since shorter and lighter women would be expected to have smaller neonates than taller and heavier women, taking maternal height and weight into account should help identify fetuses that are more likely to be constitutionally small or overgrown instead of being erroneously labeled as not aligned with their growth potential [37]. While country (as a proxy for local ethnic mix) has been found to be the principal factor in predicting adverse outcomes in infants compared with customizing for additional individual characteristics, there is increased recognition that customizing for race/ethnicity might have unintended clinical consequences [38, 39]. Birthweight is also known to increase with increasing parity until parity 4, with the largest increase between parity 0 and 1 (68 g on average) [40]. Male neonates weigh larger than females, an average 141 g larger at 40 weeks of gestation [21]. However, the influence of maternal short stature and nulliparity on perinatal mortality has been found to be mediated in part through

SGA indicating that smaller EFW associated with maternal constraint is both physiological and pathological [41]. Finally, other factors can influence fetal growth, such as genetic and external factors, including altitude, diet and lifestyle, and other environmental conditions beyond the six factors included in the customization profile that are often routinely and easily obtained during the antenatal period [29, 42–47].

Our study only found incremental improvements in detection of rates of neonatal morbidity and mortality at term with SGA and LGA defined by all three customization models compared to population based birthweight reference, with no difference in predictive ability (i.e., similar c-statistics across the models) which may have been due to smaller numbers of adverse outcomes in a healthier population initially recruited for the primary study goals to create a fetal growth standard [18]. However, the ability to test the customized methods in the CSL, a large pregnancy cohort, with consistent results as our smaller ultrasound study strengthens our findings. A major strength of our study was the longitudinal collection of ultrasound fetal measurements which allowed us to evaluate the effect of the six customization characteristics across gestation, and also the ability to explore not only birthweight but EFW which is arguably more important clinically when considering obstetrical interventions such as antenatal monitoring and earlier delivery to prevent stillbirth and birth related complications.

The concept of considering maternal and fetal characteristics is appealing as a personalized medicine approach, although there is controversy on whether customization for maternal and fetal factors improves clinically useful detection of SGA and LGA [6, 7]. Yet, the incremental improvement depends on several factors and the obstetric implications of customization have been understudied [8]. All three of the customization methods and the population-based birthweight reference had poor discrimination ability to predict neonatal morbidity and mortality indicating that we need to move beyond using a percentile cut-point to identify fetuses at risk even though this remains standard practice. Similarly, use of percentile cut-points to identify SGA and LGA is also ingrained in standard care, and customization is used in clinical practice [9]. We found that a customizing heteroscedastic model that allows for unstable variance in the customization characteristics may represent an incremental improvement over current customization methods in current use. Future work may consider additional maternal, fetal, and paternal factors and identify other factors related to neonatal morbidity and mortality. Randomized clinical trials are ultimately needed to compare whether and which customized chart is associated with reductions in short and long-term neonatal morbidity.

## Supporting information

**S1 Fig. Flowchart for study participants included in analysis.** From the NICHD Fetal Growth Studies–Singletons.
(DOCX)

**S2 Fig. Residuals from the model fits.** The residuals from the model fits for the Gardosi and the heteroscedastic models did not show any glaring departure from normality. The quantile regression does not assume normality and hence it was not included.
(EPS)

**S1 Table. Equations to calculate the percentiles for the Gardosi, heteroscedastic and quantile regression models.**
(DOCX)

**S2 Table. Model characteristics for Pairs of weeks.**
(XLSX)

**S3 Table. Customization coefficients for customized three models with the addition of paternal height and weight.**
(DOCX)

**S4 Table. Comparison of model performance for the three different methods in detecting SGA and LGA with morbidity in the NICHD Fetal Growth Studies–Singletons (N = 2,288).**
(DOCX)

## Acknowledgments

Institutions in the NICHD Fetal Growth Studies–Singletons include, in alphabetical order:

Christiana Care Health Systems, Newark, DE; Columbia University Medical Center, New York, NY; Fountain Valley Regional Medical Center, Fountain Valley, CA; Long Beach Memorial Medical Center, Long Beach, CA; Medical University of South Carolina, Charleston, SC; New York Hospital Queens, Flushing, NY; Northwestern University Feinburg School of Medicine, Chicago, IL; Saint Peters University Hospital, New Brunswick, NJ; The Emmes Corporation, Rockville, MD (Data coordinating center); Tufts University, Boston, MA; University of Alabama, Birmingham, AL; University of California, Irvine, Medical Center, Orange, CA; Women and Infants Hospital of Rhode Island, Providence, RI.

Institutions involved in the Consortium on Safe Labor study include, in alphabetical order: Baystate Medical Center, Springfield, MA; Cedars-Sinai Medical Center Burnes Allen Research Center, Los Angeles, CA; Christiana Care Health System, Newark, DE; Georgetown University Hospital, MedStar Health, Washington, DC; Indiana University Clarian Health, Indianapolis, IN; Intermountain Healthcare and the University of Utah, Salt Lake City, Utah; Maimonides Medical Center, Brooklyn, NY; MetroHealth Medical Center, Cleveland, OH.; Summa Health System, Akron City Hospital, Akron, OH; The EMMES Corporation, Rockville MD (Data Coordinating Center); University of Illinois at Chicago, Chicago, IL; University of Miami, Miami, FL; and University of Texas Health Science Center at Houston, Houston, Texas.

## Author Contributions

**Conceptualization:** Katherine L. Grantz, Anindya Roy.

**Data curation:** Katherine L. Grantz, Dian He.

**Formal analysis:** Dian He.

**Funding acquisition:** John Owen, Daniel Skupski.

**Investigation:** Katherine L. Grantz, Stefanie N. Hinkle, John Owen, Daniel Skupski, Cuilin Zhang.

**Methodology:** Anindya Roy.

**Project administration:** Katherine L. Grantz.

**Resources:** Katherine L. Grantz.

**Software:** Katherine L. Grantz, Dian He.

**Supervision:** Katherine L. Grantz, Anindya Roy.

**Visualization:** Dian He.

**Writing – original draft:** Katherine L. Grantz, Anindya Roy.

**Writing – review & editing:** Stefanie N. Hinkle, Dian He, John Owen, Daniel Skupski, Cuilin Zhang, Anindya Roy.

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
