## [Decision Letter · Decision Letter 0]

23 Nov 2022

PONE-D-22-23564A new method for customized fetal growth reference percentilesPLOS ONE

Dear Dr. Grantz,

Thank you for submitting your manuscript to PLOS ONE. After careful consideration, we feel that it has merit but does not fully meet PLOS ONE’s publication criteria as it currently stands. Therefore, we invite you to submit a revised version of the manuscript that addresses the points raised during the review process.

We look forward to receiving your revised manuscript.

Kind regards,

Quetzal A. Class, PhD

Academic Editor

PLOS ONE

Journal Requirements

2. Thank you for including your ethics statement:  "IRB approval was obtained at all institutions as well as the NIH #09-CH-N152 on December 2009. Participants gave written informed consent prior to collection of data."

For studies reporting research involving human participants, PLOS ONE requires authors to confirm that this specific study was reviewed and approved by an institutional review board (ethics committee) before the study began. Please provide the specific name of the ethics committee/IRB that approved your study, or explain why you did not seek approval in this case.

This research was supported, in part, by the Division of Population Health Research, Division of Intramural Research, Eunice Kennedy Shriver National Institute of Child Health and Human Development (NICHD), National Institutes of Health; and, in part, with Federal funds for the NICHD Fetal Growth Studies – Singletons (Contract Numbers:  HHSN275200800013C; HHSN275200800002I; HHSN27500006; HHSN275200800003IC; HHSN275200800014C; HHSN275200800012C; HHSN275200800028C; HHSN275201000009C); and in part, with Federal funds for the Consortium on Safe Labor (Contract Number HHSN267200603425C). KLG has contributed to this work as part of her official duties as an employee of the United States Federal Government.

This research was supported, in part, by the Division of Population Health Research, Division of Intramural Research, Eunice Kennedy Shriver National Institute of Child Health and Human Development (NICHD), National Institutes of Health; and, in part, with Federal funds for the NICHD Fetal Growth Studies – Singletons (Contract Numbers:  HHSN275200800013C; HHSN275200800002I; HHSN27500006; HHSN275200800003IC; HHSN275200800014C; HHSN275200800012C; HHSN275200800028C; HHSN275201000009C); and in part, with Federal funds for the Consortium on Safe Labor (Contract Number HHSN267200603425C). KLG has contributed to this work as part of her official duties as an employee of the United States Federal Government.

However, funding information should not appear in the Acknowledgments section or other areas of your manuscript. We will only publish funding information present in the Funding Statement section of the online submission form. 

This research was supported, in part, by the Division of Population Health Research, Division of Intramural Research, Eunice Kennedy Shriver National Institute of Child Health and Human Development (NICHD), National Institutes of Health; and, in part, with Federal funds for the NICHD Fetal Growth Studies – Singletons (Contract Numbers:  HHSN275200800013C; HHSN275200800002I; HHSN27500006; HHSN275200800003IC; HHSN275200800014C; HHSN275200800012C; HHSN275200800028C; HHSN275201000009C); and in part, with Federal funds for the Consortium on Safe Labor (Contract Number HHSN267200603425C). KLG has contributed to this work as part of her official duties as an employee of the United States Federal Government.

5. We noted in your submission details that a portion of your manuscript may have been presented or published elsewhere. Please clarify whether this conference proceeding or publication was peer-reviewed and formally published. If this work was previously peer-reviewed and published, in the cover letter please provide the reason that this work does not constitute dual publication and should be included in the current manuscript.

6. We note that you have indicated that data from this study are available upon request. PLOS only allows data to be available upon request if there are legal or ethical restrictions on sharing data publicly. For more information on unacceptable data access restrictions, please see http://journals.plos.org/plosone/s/data-availability#loc-unacceptable-data-access-restrictions. 

7. We note that you have stated that you will provide repository information for your data at acceptance. Should your manuscript be accepted for publication, we will hold it until you provide the relevant accession numbers or DOIs necessary to access your data. If you wish to make changes to your Data Availability statement, please describe these changes in your cover letter and we will update your Data Availability statement to reflect the information you provide.

8. We note that you have included the phrase “data not shown” in your manuscript. Unfortunately, this does not meet our data sharing requirements. PLOS does not permit references to inaccessible data. We require that authors provide all relevant data within the paper, Supporting Information files, or in an acceptable, public repository. Please add a citation to support this phrase or upload the data that corresponds with these findings to a stable repository (such as Figshare or Dryad) and provide and URLs, DOIs, or accession numbers that may be used to access these data. Or, if the data are not a core part of the research being presented in your study, we ask that you remove the phrase that refers to these data.

9. Please include your full ethics statement in the ‘Methods’ section of your manuscript file. In your statement, please include the full name of the IRB or ethics committee who approved or waived your study, as well as whether or not you obtained informed written or verbal consent. If consent was waived for your study, please include this information in your statement as well. 

10. Please include captions for your Supporting Information files at the end of your manuscript, and update any in-text citations to match accordingly. Please see our Supporting Information guidelines for more information: http://journals.plos.org/plosone/s/supporting-information. 

Additional Editor Comments:

Dear Author,

Thank you for your contribution. We continue to assert that your work is important, though the reviewers and I believe that it would be stronger with some further thoughts and considerations as well as some qualification in your claims. Please see both reviewers' comments and address.

Reviewers' comments:

Reviewer's Responses to Questions

**Comments to the Author**

1. Is the manuscript technically sound, and do the data support the conclusions?

Reviewer #1: Yes

Reviewer #2: Yes

2. Has the statistical analysis been performed appropriately and rigorously? 

Reviewer #1: I Don't Know

Reviewer #2: Yes

3. Have the authors made all data underlying the findings in their manuscript fully available?

Reviewer #1: Yes

Reviewer #2: No

4. Is the manuscript presented in an intelligible fashion and written in standard English?

Reviewer #1: Yes

Reviewer #2: No

5. Review Comments to the Author

Reviewer #1: In this study Authors constructed a new model for predicting fetal growth, The subject is of interest, the ms well written so I would like to congratulate with Authors for their effort

My comments are

1)since many of the reader are not familirar with heteroscedastic model I suggest to add an explanation of the rationale of using this approach and how it was constructed

2)of interest it should be the % variation at each gestational age in the comparison of the 3 models and if differences are significant

3) i will consider in the reference list and in the discussion the paper of Ghi et al JUM 2016 that similarly also used father characteristics in the construction of the customized mode

Reviewer #2: The authors use longitudinal fetal weight measurements they obtained in the NICHD Fetal Growth Studies to assess some of the assumptions of the Gardosi’s customized chart which assumes that i) standard deviation is a fixed proportion of the mean, and that ii) the effect of maternal characteristics is proportional throughout gestation and affects all percentiles of the distribution. Specifically, the authors evaluated the normality and constant coefficient of variation (CV) assumptions and propose the hetroscedastic and quantile regressions based percentiles as an alternative.

Major:

1) Although the attention to the non-normality of the fetal growth data is warranted, the WHO and PRB/NICHD standards already proposed the use of quantile regression. Moreover, the PRB/NICHD standard questioned the proportionality of the effects of maternal characteristics and showed differential effects on specific percentiles. The paper should put these results in perspective and perhaps somehow tone down claims of novelty.

2) Comparison of customized and non-customized standards were also evaluated for prediction of adverse outcomes and the authors miss to acknowledge and discuss previous work (https://pubmed.ncbi.nlm.nih.gov/31006913/).

3) Studying the behavior of the mean and SD only answers questions about the constant CV, but not about the normality. This should better emphasized the paper.

4. For the assessment of normality, the authors compare the Gardosi percentile with the percentiles generated empirically (for each sub-interval). This does not really answer the question, as the difference between the two values may also be the result of violation of the constant CV (proportionality) assumption. Other contributing factors could be the different datasets involved (Gardosi uses all the avalble data set; empirical uses a dataset in a given sub-interval).

5. In Fig 1, we the black line (and dots) for Gardosi model are missing but shown in the legend.

6. The authors noted that “In Fig 2…. If the Gardosi assumption of constant CV was satisfied the standard deviation would fall on the line with constant slope.” This can be true provided that data points in each interval are sufficiently large. It’s important that the authors also give number of data points in each interval.

6. PLOS authors have the option to publish the peer review history of their article (what does this mean?). If published, this will include your full peer review and any attached files.

Reviewer #1: **Yes: **Giuseppe Rizzo

Reviewer #2: No

---

## [Author Response · Author response to Decision Letter 0]

23 Jan 2023

Note- response to the reviewer comments were also provided in a Word document in the Cover Letter.

Editor Comments:

Dear Author,

Thank you for your contribution. We continue to assert that your work is important, though the reviewers and I believe that it would be stronger with some further thoughts and considerations as well as some qualification in your claims. Please see both reviewers' comments and address.

Response: Thank you for the opportunity to revise the manuscript. We have addressed the reviewers comments as outlined below.

Reviewers' comments:

Reviewer's Responses to Questions

Comments to the Author

1. Is the manuscript technically sound, and do the data support the conclusions?

Reviewer #1: Yes

Reviewer #2: Yes

2. Has the statistical analysis been performed appropriately and rigorously? 

Reviewer #1: I Don't Know

Reviewer #2: Yes

3. Have the authors made all data underlying the findings in their manuscript fully available?

The PLOS ONE data policy requires authors to make all data underlying the findings described in their manuscript fully available without restriction, with rare exception (please refer to the Data Availability Statement in the manuscript PDF file). The data should be provided as part of the manuscript or its supporting information, or deposited to a public repository. For example, in addition to summary statistics, the data points behind means, medians and variance measures should be available. If there are restrictions on publicly sharing data—e.g. participant privacy or use of data from a third party—those must be specified.

Reviewer #1: Yes

Reviewer #2: No

4. Is the manuscript presented in an intelligible fashion and written in standard English?

Reviewer #1: Yes

Reviewer #2: No

5. Review Comments to the Author

Reviewer #1: In this study Authors constructed a new model for predicting fetal growth, The subject is of interest, the ms well written so I would like to congratulate with Authors for their effort

My comments are

1) since many of the reader are not familirar with heteroscedastic model I suggest to add an explanation of the rationale of using this approach and how it was constructed

Response: The Gardosi method constructs the percentile cutpoints to define SGA and LGA, namely, 10th and 90th, in a specific way. The method is based on the assumption that the coefficient of variation (CV) is constant. Our first objective was to investigate this assumption. For the second objective, the advantage of creating a new method for customization using a heteroscedastic model is that it directly calculates the CV based on the empiric data without making such assumption. Therefore, the heteroscedastic model allows for more flexibility. We have clarified this description in the Introduction (new text underlined) (Pages 4-5):

“Second, we developed an expanded, alternative TOW customization method for estimating we created a new customization method that has more flexibility in calculating customized percentiles using a heteroscedastic regression that customizes both customized mean TOW (and hence EFW by extrapolation) and customized standard deviation, leading to customized percentiles, using a heteroscedastic regression.[16] To be precise, the heteroscedastic model customizes a transformed value of the standard deviation but because that makes the standard deviation depend on the customizing factors, hereafter we refer to it as a model for customizing the standard deviation.”

2) of interest it should be the % variation at each gestational age in the comparison of the 3 models and if differences are significant

Response: Thank you for this suggestion. We revised Figure 3 to add confidence bands for the 10th and 90th percentiles for each week. (new Fig 3) We also provide the percentiles by week to present the absolute differences and calculated the relative differences in the Table below:

GA Range of 90th and 10th in Gardosi Range of 90th and 10th in Hetero Range of 90th and 10th in Quantile Relative difference between Gardosi and Heteroscedastic a

% Relative difference between Gardosi and Quantile b

%

37 959 982 962 -2.4 -0.3

38 1042 969 963 7.0 7.5

39 1116 962 948 13.8 15.1

40 1147 949 918 17.3 19.9

41 1176 937 903 20.3 23.2

42 1216 921 921 24.2 24.3

GA, gestational age; Hetero, heteroscedastic

a Relative difference was calculated using the range as [(Gardosi–Hetero)/Gardosi] *100%

b Relative difference was calculated using the range as [(Gardosi–Quantile)/Gardosi] *100%

3) i will consider in the reference list and in the discussion the paper of Ghi et al JUM 2016 that similarly also used father characteristics in the construction of the customized mode

Response: We originally omitted this study because it didn’t report EFW or birthweight which was the focus of our study; however, we have now added this reference to the discussion. Page 25, new text underlined: 

“These findings are similar to findings from the Generation R cohort of EFW in the Netherlands [34] and fetal biometric measurements in an Italian cohort [new reference 35] although…”

Reviewer #2: The authors use longitudinal fetal weight measurements they obtained in the NICHD Fetal Growth Studies to assess some of the assumptions of the Gardosi’s customized chart which assumes that i) standard deviation is a fixed proportion of the mean, and that ii) the effect of maternal characteristics is proportional throughout gestation and affects all percentiles of the distribution. Specifically, the authors evaluated the normality and constant coefficient of variation (CV) assumptions and propose the hetroscedastic and quantile regressions based percentiles as an alternative.

Major:

1) Although the attention to the non-normality of the fetal growth data is warranted, the WHO and PRB/NICHD standards already proposed the use of quantile regression. Moreover, the PRB/NICHD standard questioned the proportionality of the effects of maternal characteristics and showed differential effects on specific percentiles. The paper should put these results in perspective and perhaps somehow tone down claims of novelty.

Response: We agree with the reviewer that others have used quantile regression and questioned the proportionality assumption of the Gardosi customization method. The novelty of our analysis is the in-depth examination and explaining whether Gardosi assumptions hold. We have clarified this point in the paper and added the WHO and PRB references (new text underlined):

 “We performed an in-depth examination of the statistical assumptions of the Gardosi customization method.” (p. 23)

“ Quantile regression was used to create the WHO fetal growth charts and also for a customized fetal growth reference in an African-American population. (new references 30, 31; p. 24)

2) Comparison of customized and non-customized standards were also evaluated for prediction of adverse outcomes and the authors miss to acknowledge and discuss previous work (https://pubmed.ncbi.nlm.nih.gov/31006913/).

Response: While we do recognize the important contribution of the Kabiri paper, our goal was not to compare customized and non-customized standards, as we agree that this subject has been extensively studied. Our goal was to compare the statistical methods of customization. We have now added the Kabiri reference to the discussion (new reference 30) as outlined in response to Reviewer 2, Comment 1 above and edited the Discussion as follows (p. 24, new text underlined):

“In the study by Kabiri et al, a customized fetal growth reference based on quantile regression did not improve prediction of perinatal morbidity compared with ultrasound references. While in the present cohort the model also did not show significant improvement in terms of birthweight prediction, it is expected that as more data from controlled studies become available, the merits of flexible models compared to linear regression-based models could be better evaluated in the context of birthweight customization.”

3) Studying the behavior of the mean and SD only answers questions about the constant CV, but not about the normality. This should better emphasized the paper.

Response: The reviewer is correct that the reported evidence based on the mean and the SD and the bin-specific percentiles do not provide sufficient evidence toward normality as the reason for departure could be due to the violation of constant CV assumption. We further investigated the assumption of normality by checking the residuals from the model fits for the Gardosi and the heteroscedastic models. The quantile regression does not assume normality and hence it is not included in the investigation. The residuals do not show any glaring departure from normality. We have now added this information to the results (p. 18) and in a new s2 Fig. 

4) For the assessment of normality, the authors compare the Gardosi percentile with the percentiles generated empirically (for each sub-interval). This does not really answer the question, as the difference between the two values may also be the result of violation of the constant CV (proportionality) assumption. Other contributing factors could be the different datasets involved (Gardosi uses all the avalble data set; empirical uses a dataset in a given sub-interval).

Response: Our goal was not really to assess normal or abnormal. Rather, we wanted to examine how robust the Gardosi estimates were. We agree that it doesn’t say anything conclusive about normality which is why we checked the residuals and have now added to the paper as outlined in response to Reviewer 2, comment 3 above. 

5) In Fig 1, we the black line (and dots) for Gardosi model are missing but shown in the legend.

Response: That part of the legend was indicating that the lines are from the models while the dots are the empirical observations. We have removed that information from the figure itself to avoid confusion and added to the figure description. (p. 13) 

“The lines are from the models while the points are empirical observations.”

6) The authors noted that “In Fig 2…. If the Gardosi assumption of constant CV was satisfied the standard deviation would fall on the line with constant slope.” This can be true provided that data points in each interval are sufficiently large. It’s important that the authors also give number of data points in each interval.

Response: We have now added the sample size in each group in a footnote. (p. 13)

”The sample size of each group is 5, 25, 85, 292, 668, 822, 350, and 41 for the point representing the grouped bins from smallest to largest predicted birthweight, respectively.” Some of the points on either edge are based on fewer sample points so we also added a new line (red line) illustrating the constant coefficient variation assumption per the Gardosi model. As you can see, the bulk of the points do not fall where one would expect them to be under the Gardosi assumption. (revised Fig 2; revised legend p. 13)

6. PLOS authors have the option to publish the peer review history of their article. If published, this will include your full peer review and any attached files.

Do you want your identity to be public for this peer review? For information about this choice, including consent withdrawal, please see our Privacy Policy.

Reviewer #1: Yes: Giuseppe Rizzo

Reviewer #2: No

While revising your submission, please upload your figure files to the Preflight Analysis and Conversion Engine (PACE) digital diagnostic tool. PACE helps ensure that figures meet PLOS requirements. To use PACE, you must first register as a user. Registration is free. Then, login and navigate to the UPLOAD tab, where you will find detailed instructions on how to use the tool. If you encounter any issues or have any questions when using PACE, please email PLOS at figures@plos.org. Please note that Supporting Information files do not need this step.

Journal Requirements

Response: The manuscript has been revised according to journal style requirements.

2. Thank you for including your ethics statement: "IRB approval was obtained at all institutions as well as the NIH #09-CH-N152 on December 2009. Participants gave written informed consent prior to collection of data."

For studies reporting research involving human participants, PLOS ONE requires authors to confirm that this specific study was reviewed and approved by an institutional review board (ethics committee) before the study began. Please provide the specific name of the ethics committee/IRB that approved your study, or explain why you did not seek approval in this case.

Response: IRB approval was received before the study began. The Methods section has been edited with the information in the ethics statement (p. 5, new text underlined): 

“Institutional review board approval was obtained at all participating sites as well as the NIH (IRB approval #09-CH-N152) on December 2009 prior to the study beginning. All participants provided informed consent prior to data collection.”

This research was supported, in part, by the Division of Population Health Research, Division of Intramural Research, Eunice Kennedy Shriver National Institute of Child Health and Human Development (NICHD), National Institutes of Health; and, in part, with Federal funds for the NICHD Fetal Growth Studies – Singletons (Contract Numbers: HHSN275200800013C; HHSN275200800002I; HHSN27500006; HHSN275200800003IC; HHSN275200800014C; HHSN275200800012C; HHSN275200800028C; HHSN275201000009C); and in part, with Federal funds for the Consortium on Safe Labor (Contract Number HHSN267200603425C). KLG has contributed to this work as part of her official duties as an employee of the United States Federal Government.

Response: The Role of Funder statement has been added to the cover letter.

4. Thank you for stating the following in the Acknowledgments Section of your manuscript: This research was supported, in part, by the Division of Population Health Research, Division of Intramural Research, Eunice Kennedy Shriver National Institute of Child Health and Human Development (NICHD), National Institutes of Health; and, in part, with Federal funds for the NICHD Fetal Growth Studies – Singletons (Contract Numbers: HHSN275200800013C; HHSN275200800002I; HHSN27500006; HHSN275200800003IC; HHSN275200800014C; HHSN275200800012C; HHSN275200800028C; HHSN275201000009C); and in part, with Federal funds for the Consortium on Safe Labor (Contract Number HHSN267200603425C). KLG has contributed to this work as part of her official duties as an employee of the United States Federal Government. However, funding information should not appear in the Acknowledgments section or other areas of your manuscript. We will only publish funding information present in the Funding Statement section of the online submission form. 

Please remove any funding-related text from the manuscript and let us know how you would like to update your Funding Statement. Currently, your Funding Statement reads as follows: This research was supported, in part, by the Division of Population Health Research, Division of Intramural Research, Eunice Kennedy Shriver National Institute of Child Health and Human Development (NICHD), National Institutes of Health; and, in part, with Federal funds for the NICHD Fetal Growth Studies – Singletons (Contract Numbers: HHSN275200800013C; HHSN275200800002I; HHSN27500006; HHSN275200800003IC; HHSN275200800014C; HHSN275200800012C; HHSN275200800028C; HHSN275201000009C); and in part, with Federal funds for the Consortium on Safe Labor (Contract Number HHSN267200603425C). KLG has contributed to this work as part of her official duties as an employee of the United States Federal Government. Please include your amended statements within your cover letter; we will change the online submission form on your behalf.

Response: As federal employees, we are required to disclose funding on all published manuscripts. I have removed from the manuscript and added to the cover letter as instructed, but please make sure that this information is included with the manuscript itself.

5. We noted in your submission details that a portion of your manuscript may have been presented or published elsewhere. Please clarify whether this conference proceeding or publication was peer-reviewed and formally published. If this work was previously peer-reviewed and published, in the cover letter please provide the reason that this work does not constitute dual publication and should be included in the current manuscript.

Response: To clarify, there was no abstract formally published for this conference proceeding.

6. We note that you have indicated that data from this study are available upon request. PLOS only allows data to be available upon request if there are legal or ethical restrictions on sharing data publicly. For more information on unacceptable data access restrictions, please see http://journals.plos.org/plosone/s/data-availability#loc-unacceptable-data-access-restrictions.

a. If there are ethical or legal restrictions on sharing a de-identified data set, please explain them in detail (e.g., data contain potentially sensitive information, data are owned by a third-party organization, etc.) and who has imposed them (e.g., an ethics committee). Please also provide contact information for a data access committee, ethics committee, or other institutional body to which data requests may be sent.

b. If there are no restrictions, please upload the minimal anonymized data set necessary to replicate your study findings as either Supporting Information files or to a stable, public repository and provide us with the relevant URLs, DOIs, or accession numbers. For a list of acceptable repositories, please see http://journals.plos.org/plosone/s/data-availability#loc-recommended-repositories.

Response: We have submitted the data and accompanying files to NICHD BRADS: https://brads.nichd.nih.gov/

7. We note that you have stated that you will provide repository information for your data at acceptance. Should your manuscript be accepted for publication, we will hold it until you provide the relevant accession numbers or DOIs necessary to access your data. If you wish to make changes to your Data Availability statement, please describe these changes in your cover letter and we will update your Data Availability statement to reflect the information you provide.

Response: We have submitted the data and accompanying files to NICHD BRADS. The administrators will notify me when the work is complete (estimated mid-January) and I will notify you when the data is available.

8. We note that you have included the phrase “data not shown” in your manuscript. Unfortunately, this does not meet our data sharing requirements. PLOS does not permit references to inaccessible data. We require that authors provide all relevant data within the paper, Supporting Information files, or in an acceptable, public repository. Please add a citation to support this phrase or upload the data that corresponds with these findings to a stable repository (such as Figshare or Dryad) and provide and URLs, DOIs, or accession numbers that may be used to access these data. Or, if the data are not a core part of the research being presented in your study, we ask that you remove the phrase that refers to these data.

Response: As the data are not a core part of the research being presented in our study, we have removed this statement from the methods (p. 18).

9. Please include your full ethics statement in the ‘Methods’ section of your manuscript file. In your statement, please include the full name of the IRB or ethics committee who approved or waived your study, as well as whether or not you obtained informed written or verbal consent. If consent was waived for your study, please include this information in your statement as well. 

Response: This information has been added to the Methods section (p. 5):

“Institutional review board approval was obtained at all participating sites as well as the NIH (IRB approval #09-CH-N152) on December 2009 prior to the study beginning. All participants provided written informed consent prior to data collection.”

10. Please include captions for your Supporting Information files at the end of your manuscript, and update any in-text citations to match accordingly. Please see our Supporting Information guidelines for more information: : http://journals.plos.org/plosone/s/supporting-information

Response: Captions for our Supporting Information files have been added at the end of our manuscript. We have also updated in-text citations to match accordingly.

---

## [Decision Letter · Decision Letter 1]

23 Feb 2023

A new method for customized fetal growth reference percentiles

PONE-D-22-23564R1

Dear Dr. Grantz,

We’re pleased to inform you that your manuscript has been judged scientifically suitable for publication and will be formally accepted for publication once it meets all outstanding technical requirements.

Kind regards,

Quetzal A. Class, PhD

Academic Editor

PLOS ONE

Additional Editor Comments (optional):

Dear Authors,

Thank you for your dedicated responses to concerns raised by the reviewers and myself. There is one minor change that can be addressed during the editing process -- an incorrect citation as pointed out by reviewer 2 -- but otherwise we accept your manuscript for publication.

Thank you,

Quetzal Class

Reviewers' comments:

Reviewer's Responses to Questions

**Comments to the Author**

1. If the authors have adequately addressed your comments raised in a previous round of review and you feel that this manuscript is now acceptable for publication, you may indicate that here to bypass the “Comments to the Author” section, enter your conflict of interest statement in the “Confidential to Editor” section, and submit your "Accept" recommendation.

Reviewer #1: All comments have been addressed

Reviewer #2: All comments have been addressed

2. Is the manuscript technically sound, and do the data support the conclusions?

Reviewer #1: Yes

Reviewer #2: Yes

3. Has the statistical analysis been performed appropriately and rigorously? 

Reviewer #1: Yes

Reviewer #2: Yes

4. Have the authors made all data underlying the findings in their manuscript fully available?

Reviewer #1: Yes

Reviewer #2: Yes

5. Is the manuscript presented in an intelligible fashion and written in standard English?

Reviewer #1: Yes

Reviewer #2: Yes

6. Review Comments to the Author

Reviewer #1: The manuscript was revised successfully and al the queries were properly addressed. Congratulations !

Reviewer #2: The authors have address most of my questions. There is one minor inconsistency to address, and the authors can be entrusted to address without delaying the acceptance of the paper.

1) To my initial point #1 the authors agree to mention other growth standards that have use quantile regression and questioned the normality assumption. They cited the WHO standard and mentioned the PRB/NICHD standard but for the latter the reference is incorrect. It should be: https://pubmed.ncbi.nlm.nih.gov/29422207/. The Kabiri et al reference is for the study comparison of standards for prediction of neonatal outcomes, as mentioned in the discussion.

7. PLOS authors have the option to publish the peer review history of their article (what does this mean?). If published, this will include your full peer review and any attached files.

Reviewer #1: **Yes: **Giuseppe Rizzo

Reviewer #2: No

---

## [Editor Report · Acceptance letter]

7 Mar 2023

PONE-D-22-23564R1 

A new method for customized fetal growth reference percentiles 

Dear Dr. Grantz:

I'm pleased to inform you that your manuscript has been deemed suitable for publication in PLOS ONE. Congratulations! Your manuscript is now with our production department. 

Kind regards, 

on behalf of

Dr. Quetzal A. Class 

Academic Editor

PLOS ONE